# Challenges in Getting Started in Motion Graphic Design: Perspectives from Casual and Professional Motion Designers

Amir Jahanlou*

School of Interactive Arts and Technology, Simon Fraser University

William Odom†

School of Interactive Arts and Technology, Simon Fraser University

Parmit Chilana ‡

School of Computing Science, Simon Fraser University

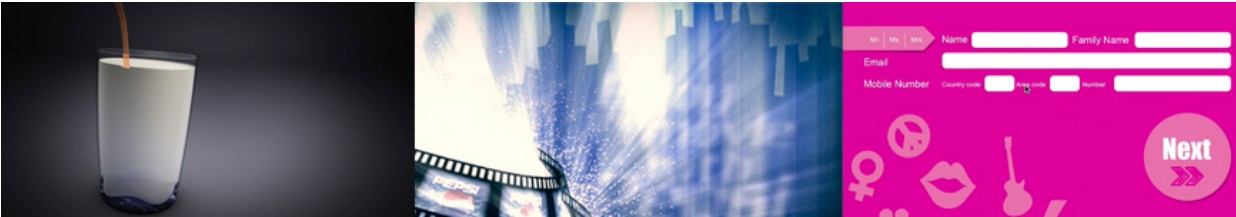

Figure 1: Examples of motion graphics in education (left), advertising (middle), and UI onboarding (right).

## ABSTRACT

Motion graphics videos offer a powerful means of communicating complex concepts through engaging visuals and animation. While filmmaking, marketing, or video games industries use these videos to tell compelling stories, others such as educators or domain experts face a steep learning curve. Creating even a short motion graphics video can be an arduous process that requires competency in scriptwriting, graphic design, animation, and skills in using various feature-rich software applications. We interviewed 19 casual and professional motion designers working on a range of motion graphics projects to understand their design processes and challenges. Our results reveal several difficulties that new motion designers face in getting started in the field and how they struggle to devise workarounds. We identify opportunities for HCI to lower entry barriers by designing user-centered tools that simplify the motion design process and incorporate example-based learning and collaborative approaches.

**Keywords**: Motion graphics, multimedia, explainer videos, infographics, motion design.

**Index Terms**: H.5.2 [Information Interfaces and Presentation]: User Interfaces; I.3.2 [Computer Graphics]: Graphics Systems.

## 1 INTRODUCTION

Motion graphics videos use a combination of sound and movement to convey ideas through computer-generated graphics, icons, and pictograms [29] (Figure 1). The phrase motion graphics emerged from graphic design in motion ('*motion design*') [24] and has a long history in film and television production [58]. With the growth of video as a communication medium and the access to video publishing platforms [2], these videos have become popular to create lively and engaging content [20,24,44] in marketing and advertising contexts. Motion graphics videos are also being used in informational contexts to illustrate complex phenomena in physics [59], geography [17], healthcare [57], and other domains [17,64].

---

\* email: amir_jahanlou@sfu.ca

† email: wodom@sfu.ca

‡ email: pchilana@cs.sfu.ca

Despite the promise of motion graphics videos in informational fields, creating such videos can be a laborious and expensive process. For example, surveys suggest that it can cost up to $8000 USD to make a short 60-second motion graphics video [69]. Unlike camera or screen-recorded videos where the captured footage serves as a starting point, the creation of motion graphics videos is an entirely software-driven process that begins with a blank canvas: the content and the movement are usually generated from scratch [22]. Furthermore, creating a motion graphics video requires scriptwriting, animation and graphics design, familiarity with feature-rich software, and the knowledge of color, typography, sound, or imagery [29]. Given the high costs and array of skills needed to create a motion graphics video, traditionally, only large organizations have been able to afford motion designers' expertise. As a result, motion graphics has primarily catered to sectors such as marketing, video games, film, or advertising. Domain experts, such as scientists or instructors (who could benefit from using this information-sharing medium), face a high barrier of entry in creating motion graphics on their own [24].

Although research in HCI and media studies has a long history of exploring visual storytelling [28,45,55], approaches to animation [46], interactive software design [15,30,50], and graphic design [10,38,62] surprisingly, not much is known about the processes and workflows used by motion designers. In this paper, we argue that there is a rich opportunity for the HCI community to better understand and facilitate the process of creating motion graphics to empower more individuals to use this medium in information-sharing contexts. This has parallels to existing efforts in HCI to increase the participation of casual users and domain experts in various technical and design tasks, such as programming [14], website creation [35,47,51], interaction design and UX [14], and generating visualizations [43].

To investigate the work of motion designers, we conducted interviews with 19 professional and casual motion designers working on projects such as explainer videos, advertisements, education, or UI animations. We focused on understanding how these designers approached the overall design process and the types of tools and techniques they used. In particular, we probed into any barriers that they faced in getting started with motion graphics, challenges in the process, or the difficulties in communicating their progress. We found that professional motion designers invested significantly more in the preparatory stages of creating a video. In contrast, casual motion designers had difficulty understanding the "big picture" of a project and often skipped pre-production steps.

Moreover, the software tools used in motion design were not necessarily developed for motion graphics as they catered to a variety of creative processes, leading to a mismatch of expectations.

This paper makes two main contributions: a) we offer empirical insights detailing the work of motion designers, including the diverse range of challenges that they face as well as their learning approaches, issues in understanding the processes, and the experiences of working with software tools; b) we interpret these findings to offer opportunities where research in HCI can lower the barriers to entry and democratize motion graphics creation. For example, there is a key opportunity for supporting the onboarding process by helping individuals understand the bigger context of an entire motion graphics project. Furthermore, we advocate that future tools should support the end-to-end process of designing motion graphics. In doing so, new opportunities would be generated to address key challenges occurring in the preliminary steps of creating motion graphics projects.

## 2 RELATED WORK

To situate our findings in the broader literature, we highlight the efficacy of using animation and graphics in different domains, how motion graphics videos fit in with the larger context of design, and insights from works in understanding the challenges of learning and working with feature-rich software applications.

### 2.1 The Efficacy of Graphics, and Animation

While motion graphics can be considered a subgenre of animation [9][13], they are yet primarily motivated by communicating a phenomenon rather than following the story of a hero [24,65]. The positive impact of animation on the memorability of content [9] enables motion graphics to visually present concepts in different domains [24], such as education [26] or geography [17].

While motion graphics is a relatively new phenomenon, decades of textbooks with illustrations enforces the notion of using imagery for better explanations. Using animation in education has been shown to simplify understanding [6], increase attention [11], and leave a lasting memory [8]. Furthermore, human reaction to motion is partly our instinct [46], making motion graphic a rich common language [33] understood across different cultures. Moreover, movement as a means of communication [46] empowers motion graphics videos to affect different levels of the message [11]. Previous studies have highlighted the positive impact of animation in learning physics [59], geography [17], dental care [57], and even increasing motivation among educators [6]. Motion graphics videos also enable a degree of storytelling that makes them an effective method of establishing common grounds [71]. Storytelling has been found useful in the education of management [53], early years of higher education [25], as well as improving knowledge interpretation [5,71], and making complex topics more approachable by introducing them sequentially [4].

Despite the illustrated effectiveness of using animation and graphics, motion graphics creation is yet a complex activity requiring tool-specific expertise [18]. Our study highlights the challenges in getting started with motion graphics and the lack of appropriate tools that support the entire creation ecosystem. More broadly, our work contributes to a trajectory of research in the HCI community aimed at studying designers (especially from non-traditional backgrounds) to better support their creative practices, processes, and experiences (e.g., [30,60,70]).

### 2.2 Motion Graphics in the Larger Context of Design

Previous studies in HCI have explored various types of design activities. For example, studies have examined the processes of doing interactive software design [15,30,37,68] or graphic design (e.g., [62]). Although these domains of design bear some similarities to motion design, there also are key differences. For example, compared to graphics design, motion graphics entail movement and far more elaborate storytelling. In that, while graphic designers work on a static image, motion designers need to complete (and relate) many different parts of the content to create compelling stories. Also, unlike interaction design, where the goal is to produce a compelling task-specific user experience for an interactive artifact, motion design has a more nuanced focus on creating a video to convey complex information or tell a story.

Perhaps more relevant to the domain of motion graphics is research on the design of video authoring tools. However, unlike camera- or screen-captured videos that primarily rely on pre-recorded video footage to begin the authoring process, motion graphic design begins with a blank canvas, and designers have to create all of the content in an entirely software-driven process. Prior work has also explored the use of automation in video authoring – for example, by creating event timelines [12] and producing storybooks from text [34,63], or simplifying the video production using recommendation [42]- and example-based video editing [52]. In contrast to these approaches, motion design is a process driven by people and their information needs (as opposed to automated content creation). Furthermore, motion graphics in informational domains is primarily concerned with capturing the nuances of subject-matter experts and imparting specific knowledge that is not achievable in automatic content creation.

Our work complements previous studies and insights into graphics design, interaction design, animation, and video authoring, and illustrates the specific challenges faced by motion designers in the process of creating videos using computer-generated graphics, sound, and movement.

### 2.3 Challenges in Using and Learning Feature-Rich Software

HCI has a long history of exploring the difficulties of using and learning feature-rich software applications (e.g., novices and 3D printing [36], prototyping approaches by beginners [21], or graphic designers using digital tools [62]). In particular, it has been shown that it is problematic for subject-matter experts and educators [18] to work with such software as it requires a substantial commitment to training on top of their day jobs [59]. As a side-effect of these difficulties in the usability of complex software [15] or the challenges in learning multiple feature-rich software applications [40], advanced tools often get ignored in their entirety [45]. On the other hand, specific tools with limited functionality also suffer from being useful for particular conditions only. As a result of such challenges, the time required to create motion graphics videos is longer than making slideshows or even video tutorials. Such issues compound the traditional challenges of integrating complex technology and pedagogy [54,66] and is another reason why the potential of motion graphics is not fully realized in non-business domains, such as education.

Our work complements the previous studies and highlights the software and non-software related barriers for creating motion graphics videos for novices. We explore the entire creation process from the perspective of motion designers (both professional and beginners). Furthermore, our study reveals insights into the difficulties that occur not only during software use but also during the preparatory steps (pre-production) in creating a compelling narrative, as well as challenges that the novices face in understanding the context of the production. We suggest ways in which the process can be simplified for casual motion designers.

| P# | Training | Domain | Gender | Location | Education | Expertise |
|---|---|---|---|---|---|---|
| 1 | Design | Product Marketing | Female | Australia | Undergraduate | Professional |
| 2 | Self-taught | Online Advertising | Male | Brazil | Some College | Professional |
| 3 | Design | Advertising and | Male | Brazil | Undergraduate | Professional |
| 4 | Marketing | User Interface Design | Male | Ukraine | Undergraduate | Casual |
| 5 | Self-taught | Product Advertising | Female | USA | Some College | Professional |
| 6 | Science | Medical Research | Male | Colombia | Undergraduate | Casual |
| 7 | Animation | Online Advertising | Male | Singapore | Some College | Professional |
| 8 | Science | Fitness Content | Male | Canada | Undergraduate | Casual |
| 9 | Design | Product Advertising | Female | USA | Undergraduate | Professional |
| 10 | Filmmaking | Music Video | Female | USA | Undergraduate | Professional |
| 11 | Communication | Various Advertising | Male | Australia | Undergraduate | Professional |
| 12 | Filmmaking | Medical Content | Male | Brazil | Undergraduate | Casual |
| 13 | Animation | Product Marketing | Male | Germany | Undergraduate | Professional |
| 14 | Self-taught | Product Marketing | Male | Lithuania | Some College | Professional |
| 15 | Science | Astrophysics | Male | Canada | Graduate | Casual |
| 16 | Self-taught | Product Advertising | Male | USA | Some College | Professional |
| 17 | Design | Documentaries | Female | Canada | Undergraduate | Casual |
| 18 | Self-taught | Product Advertising | Male | Greece | Some College | Professional |
| 19 | Education | Health Education | Male | USA | Graduate | Casual |

Table 1: Overview of participants in our interviews.

## 3 METHOD

To better understand the practice of motion graphics design and motion designers' work, we carried out interviews with a range of motion designers. Below, we present our research approach, details of recruiting participants who represented a range of motivations and skills, and our data analysis strategies.

### 3.1 Semi-Structured Interviews with Motion Designers

We carried out semi-structured interviews with motion designers from a variety of backgrounds and levels of expertise. We recruited 19 motion designers (14 males, 5 females summarized in Table 1) with a range of expertise (2 to 12 years of experience) through searching for online motion graphics videos on various video publishing websites (e.g., vimeo.com [67]). In doing so, we searched for recently published explainer or motion graphics videos and reached out to the creators, and invited them for an interview. When possible, we conducted face-to-face interviews, and for remote participants, we completed the interviews through Skype. Each interview lasted about 45 minutes, and participants received a $15 Amazon gift card for their participation.

Before starting the interview, we used a questionnaire to collect demographic information such as gender, age, occupation, education, experiences with motion graphics, and motivation for working with motion graphics. We then started the interview with warm-up questions asking participants to recall their most recent project and encouraged them to describe their typical process and any challenges that they faced. We then sought to understand the designers' processes by asking questions about their approach to learning new techniques, workarounds, and the types of resources they consulted to support their practice. We wanted to understand what kinds of situations triggered the need for particular kinds of resources, how they could locate them, and how useful they found these different resources. We used common resources to prompt the participants (e.g., online courses, books, tutorials, in-person training, etc.). After the first few interviews, we updated this list with more informal resources that have been mentioned, such as blogs, magazines, or forums.

During our interviews, we observed that most participants (12/19) were professional motion designers in that they had sufficient training and routinely worked on commercial projects. However, we also saw a class of participants (7/19) that could be described as casual motion designers working on a single project and mostly new to the field. This realization encouraged us to prompt all participants to reflect back on their early days of working with motion graphics and speak of the challenges they encountered in getting started. These insights helped us better understand the differences in casual vs. professional motion designers' perspectives and the prevalence of the key themes from our interviews across a larger pool of motion designers from different domains, motivations, and expertise. Lastly, we ended the interview by a conversation about the trends, tools, and the future of motion graphics design. In doing that, we sought to learn about motion designers' pain points and any solutions they had devised to resolve them. These insights support us in synthesizing the challenges and solutions into design implications and guidelines.

### 3.2 Data Analysis

All interview recordings were transcribed and coded using *Atlas.TI* data analysis software. The coded data were analyzed to illustrate the different processes, challenges in learning or software interactions, and the workarounds that motion designers had devised to navigate their field. We used an inductive analysis [19] approach to explore the themes around our main research question. To ensure the coded data's validity, the primary author performed the first open coding pass and consulted with another researcher to discuss and develop an initial list of codes. The initial pool included codes such as processes, pre-production, UI, and motivation. All of the researchers had prior experience with qualitative analysis and continually checked the legitimacy of the scheme. Upon completing the first phase, three researchers collectively discussed the emerging themes and finalized the coding scheme.

## 4 RESULTS: KEY INSIGHTS FROM MOTION DESIGNERS

In this section, we present the key results that we synthesized from our interviews, highlighting the diversity in the backgrounds of motion designers and their projects, the challenges they face during the design process, how they approach learning, and the role of collaboration and online communities.

### 4.1 Challenges in Using Motion Graphics Across Different Domains

Our participants (summarized in Table 1) worked in various domains, such as product marketing, user interface design, fitness

content creation, music video production, astrophysics, health educational videos, and documentary filmmaking. Those classified as professionals (12/19) often had formal training in creative domains such as animation, design, or filmmaking. In contrast, casual motion designers (7/12) -who did not have formal training in design or animation- learned motion graphics to meet specific demands of their domains (e.g., to create explainer videos, educational videos, or physical fitness). Furthermore, while some participants had the option to solicit professional motion graphics services, this was costly, and the end-product did not accurately reflect the subtleties of the subject matter.

The diversity of content forced both casual and professional motion designers to continuously learn new techniques or regularly create new and unfamiliar content. For example, P12 working on medical field projects described how they had to learn to create artworks that were specific to a particular domain to illustrate various procedures. Another participant who worked on educational projects explained how they had to calibrate their content's complexity depending on the target audience:

*If it's for kids… I would just try something simpler. So, I will try to use many colors… For Science, I can go as complicated as I want with motion graphics because the audience is [scientists] who already know the subject… I can put all these weird names that for me don't make sense, and no one will get it. But, these people [scientists], this special audience will understand it. (P19)*

In addition to participants who worked on informational content, some individuals worked on more creative productions, such as exploring motion graphics for music videos, which added to the complexity of interpreting the briefs and coming up with relevant content and animation:

*I am contacted by independent artists, mainly musicians who ask me to animate their music videos or things like that. Or some people such as independent filmmakers ask: "Can you animate this?". (P10)*

Despite the challenges in navigating a diverse field, we found that all motion designers in our study enjoyed the freedom to create motion graphics independently and to express a variety of ideas using motion graphics in their "own voice":

*With motion graphics, I guess my main idea is that you can experiment a lot and that was really appealing whereas if you go video production, you [have] very little creative freedom and I don't really like. So, my thinking was that I want to be more independent and with the motion graphics, it offered that option. (P17)*

## 4.2 Challenges in Creating a Motion Graphics Video

Video production can typically be described as a process with three stages of pre-production, production, and post-production [49]. We use these phases as a lens to analyze the process of creating a motion graphics video and highlight a range of practices and challenges that participants faced across different stages. Each phase of the production process includes multiple steps that add to the complexity of the process for non-experts (Figure 2). Furthermore, despite the seemingly sequential nature of the process, motion designers often had to work on several steps at the same time and equip themselves with the know-how of adequately allocating time to each stage or reducing the length of others.

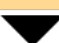
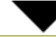
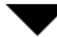
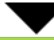

Figure 2: A typical process of creating a motion graphics video and the iteration between steps (e.g., style frame, mood-boards, etc.). Casual motion designers were often unaware of the significance of these steps and skipped most aspects of pre-production; in contrast, professional motion designers spent most of their time in this stage.

### 4.2.1 Pre-Production Challenges

Pre-production supports an understanding of the project's scope, which results in better planning for the video [32]. During pre-production, motion designers usually convert the project's brief to a script, create a storyboard, prepare a mood-board, or edit animatics (draft videos). However, most participants perceived this process to be time-consuming and expressed that they could not initially appreciate how their work benefited from pre-production. As a result, casual motion designers would often skip the pre-production steps and prematurely begin working on the final video, only having to stop the production later and re-start the process:

*[Beginners] usually skip [pre-production] phase, and that makes them spend much more time on the animation stage. Because they don't want to plan. They open the software and start an animation. (P3)*

Over time, motion designers had learned to invest more in pre-production. The first step usually was developing a brief for the project and converting that into a script. Participants spoke of a myriad of techniques for writing the scripts. As shown in other works [7,49], the script included details such as action and the voice over the videos. However, scripts were not always easy to create. A participant working on educational content pointed out how they needed to hire someone else to write it:

*Most of the time if I'm building the script, it's not really simple, I need to hire somebody to write it. Just so it has everything fit. So, writing the script and finding the visual is the first challenge. (P16)*

Motion designers used the script to break down the video into separate shots. This breakdown helped them work on each shot independently while keeping the overall project in mind. Among the key factors for producing a shot-by-shot was the use of language structures such as sentences or paragraphs. These shot breakdowns supported the more experienced motion designers to create separate shots and edit them back together. However, motion designers expressed two challenges with creating these breakdowns; a) the first one was that there is no approach for breaking a long script into separate shots within the current software tools, and b) how existing tools did not provide a method of placing each shot within the context of an entire video. Such lack of context was rather challenging for casual motion designers as they had difficulty imagining the final video from separate shots.

While the script usually provides a scene breakdown, storyboards break up the script into key moments in time [61] using visual frames. Professionals spoke of how they resorted to basic mediums such as paper and pencil to draw low-fidelity storyboards that convey the necessary information. Casual motion designers, on the other hand, felt like they should create advanced storyboards. This attitude, coupled with their lack of skills, meant that, at times, they did not feel confident in their storyboards. Professional motion designers were mainly concerned about conveying their ideas through the storyboards. They expressed how doing that makes it easier to adjust in future steps:

*My storyboards are embarrassingly low fidelity. If I draw a circle and an arrow and I make a note at the bottom of what it's going to be, I'll know, but I don't bother fleshing them out too much because I like to keep that part of the process a little broad. (P9)*

### 4.2.2 Production and Post-Production Challenges

The interdisciplinary nature of the process (graphics, text, images, sound, movement, editing, etc.) means that several software applications are needed to create a motion graphics video. Participants, notably casual motion designers, expressed their frustration in learning many (often complex and feature-rich) tools. Our analysis revealed that at least two distinct software tools (*Adobe Photoshop* [72] and *Adobe After Effects* [16]) were used to create a motion graphics video. While participants benefited from transferring their knowledge from one software to another, even professional motion designers expressed the difficulty of working with many tools. For example, a participant working on UI videos shared their frustration:

*... it creates a lot of difficulties when you switch between software. But I know that there is no perfect software. So, each software has its own strong parts like there are advantages and disadvantages because we have to use a lot of software. (P4)*

Furthermore, to make the video's motion, all (or some) of the artwork had to be animated. Animating the artworks is a software-based process that is a primary difference between camera recorded and motion graphics videos. While in camera-recorded videos, the motion is created using actors and camera movements, in motion graphics, designers invest their time in creating and animating graphical elements in software applications. This step was described as being "*labor-intensive*" and "*time-consuming*."

Beyond the difficulty of animating, while all participants mentioned that knowing the animation principles had little to no bearing on their ability to create appealing videos, they acknowledged that the current tools are not intuitive for individuals from non-animation domains. Participants expressed how the underlying terminologies were challenging to understand for those from fields other than animation or design:

*The hardest thing for beginners [is] that they don't understand the interface from the first sight because it's not user-friendly... And there are no tips. No easy ways to tell about this software, to tell about this interface in a really quick way. (P8)*

Moreover, participants described how the state-of-the-art tools had been appropriated from other creative domains and were not necessarily designed for motion graphics workflows. As a result, motion designers had to devise a range of workarounds (e.g., macros and scripts, copying keyframes, external plugins, and re-using previous animations) for simplifying their creation process. Some had even invested time in learning programming to develop their own solutions. They expressed how using these third-party plugins was necessary for having a functioning software:

*... without extensions, scripts and plugins, After Effects wouldn't be half as powerful ... It becomes a requirement for people to know those plugins in order to be able to work in the industry and the higher league. (P13)*

While participants liked the control over their message, they also spoke of challenges with the size of the process. For instance, they expressed difficulties in choosing from many motion graphics styles or the need for a working knowledge of different software tools to create different content types:

*Motion graphics is a very broad thing. For example, character animation or visual effects... So, I hope that there is software that will be readily understood for [everyone]. Because we do not know the terms for visual effects or animation. We are not specialized in it, and we hope that there's a software that can make it easier. (P7)*

Finally, as most software tools were designed for those with training in animation or design, it was tricky for casual motion designers to begin with the software tools on their own. The lack of such DIY tools widens the gap between where video publishing has reached (where non-experts can easily publish their videos) and motion graphics creation (which is still an expert-driven activity).

### 4.3 Challenges in Learning and Locating Examples

Learning the techniques of motion graphics can be a challenging task for new motion designers. While numerous online resources offer training materials, our participants indicated that the relevant instructions and tutorials were not always easy to locate. Furthermore, we learned that casual designers often had difficulty understanding where to even begin or select appropriate techniques for their own projects. A recurring theme in the interviews was that participants usually learned the skills by watching motion graphics videos created by other designers. All participants identified *YouTube* as a key resource for learning. They further noted the importance of viewing complete projects and cited the benefits of watching "making-of" videos in which the process of creating a video is demonstrated. Participants explained how watching such making-of videos was more useful than click-and-follow tutorials as it helped them view the workflow of other designers:

*I mean if you watch a lot of motion graphics videos, you kind of get that sense for how other people do it too, and then, getting inspired from those making the scene. (P11)*

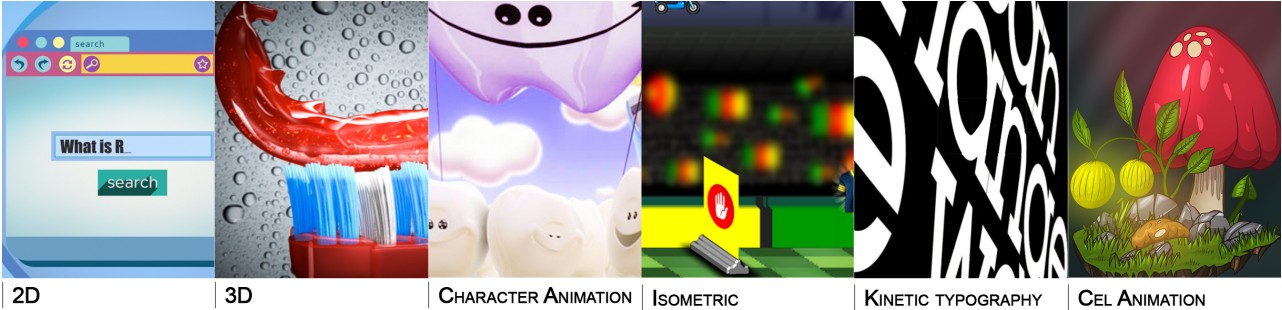

| 2D | 3D | CHARACTER ANIMATION | ISOMETRIC | KINETIC TYPOGRAPHY | CEL ANIMATION |

Figure 3: Examples of popular motion graphics styles exercised by participant. (2D: using two-dimensional graphics; 3D: using three-dimensional graphics; Character Animation: including human, creature, animal or fantasy creatures; Isometric: representing the view in a 2.5-dimensional perspective; Kinetic Typography: primarily driven by moving text; Cell Animation: that are hand-drawn animated objects).

Despite the benefits of watching *making-of* videos, we learned that casual motion designers usually struggled to understand the overall design process. Casual motion designers often had limited experience with the software applications shown in the demonstrations and the video authors sometimes assumed more advanced knowledge, which in turn resulted in casual designers being unable to apply the instructions to their needs:

*Even [if] the examples did exist, finding the precise thing you want is harder than just making it. How do I use it, I want something to fly in from the left to the right and jiggle and fly off the screen? You can't find the specific thing you need... And these issues waste a lot of time. (P6)*

Furthermore, participants also spoke of the importance of looking at examples to select the style for their videos. The style referred to the video's presentation mode, such as two- or three-dimensional (Figure 3). Each style would need a different set of skills and software applications, and some styles were more demanding to create than others (e.g., 3D was more complex and laborious than 2D). While professional motion designers could infer the complexity of either style and assess the time (and skills) required to create a video in either style, casual motion designers found this challenging. They struggled to understand the technical difficulties of making videos in each style or even the software tools required for them:

*...there [are] definitely many styles and artistic types of motion graphics. I know that that's only two categories, and it's very not descriptive, but it's ... that's a very hard task I think any motion graphics artist would have a difficult time coming up with plans for the right one [style] and the software he needs. (P10)*

Participants pointed out that a trend in motion graphics was to use premade content, such as graphical artwork, visual effects, or animated artifacts used directly within new video productions. The access to such premade content was essential for casual motion designers as they struggled with creating pieces of the artworks on their own. Furthermore, while casual motion designers relied on such premade content to feel confident and overcome the issue of the "blank canvas," the software tools alone did not provide sufficient artworks or examples. The presence of such premade content (e.g., icons, stock imagery, animated objects, etc.) would support designers to speed their process up and skip the parts that were more difficult to understand:

*The actual animating is definitely the most labor-intensive part. I would say, for creating that stuff, I think the way to speed that up [process] is to use premade templates and premade graphics. (P9)*

However, such premade content came with its own challenges. Motion designers spoke of various hurdles such as the cost associated with such content, the ability to edit them, and the challenges with locating the appropriate content within numerous providers, websites, and software:

*If there were templates that fit everything, how do you search for that? It's virtually impossible to find what you want because the space is so big. (P15)*

## 4.4 Challenges in Communicating and Collaborating

The motion designers in our study repeatedly identified the importance of visual communication as part of motion graphics creation. For these designers, communicating with stakeholders such as colleagues, managers, and clients was an important part of the job. Most of our participants (11/19) usually had to work with stakeholders remotely. They expressed many challenges in communicating the visual aspects of the artifacts through traditional communication means (i.e., text, phone, or email) as they are not designed to convey visual content:

*It's email, or the worst is the comments added a Word document that gets emailed around. That still drives me crazy. I'll still deal with studios that will send me a marked-up Word document, and I'm like, "How do you guys function?" People seem very hesitant to transition to collaborative tools. (P16)*

Motion designers resorted to approaches such as video calls with screen-sharing, cloud spaces, or even using storage disks to address issues in establishing a common ground. The introduction of new tools such as video annotation software *Frame.IO [27]* helped simplify some parts of the process. However, participants expressed concerns over using such software as their tool-set of creating motion graphics was already crowded, and every new tool presented a steep learning curve and more management:

*In interacting with the client, I use different software to get their feedback, mainly Google drive, and Frame.io. But those two may get confusing. But not one software does everything that I would like it to have. (P5)*

| | |
|---|---|
| 1 - | Motion designers come from many different backgrounds and often struggle with feature-rich software tools that assume previous training in design or animation. |
| 2 - | There are multiple different stages (e.g., script-writing, animating) in creating a motion graphics video. Motion designers (particularly casuals) struggle to understand the role of each step of the process. Furthermore, none of the current tools provides a method of contextualizing this process. |
| 3 - | Current software tools do not directly support the pre-production stage of creating motion graphics videos, which many professional motion designers believe is the most critical phase in creating a video. |
| 4 - | Motion design requires access to large repositories of examples and references (images or videos). However, novices struggle to locate relevant content and adjust them to the needs of their projects. |
| 5 - | Motion designers find it most useful to look at examples and relevant step-by-step instructions, but most current tools lack such example-based learning opportunities for creating motion graphics. |
| 6 - | Multiple tools are required to create even a short motion graphics video, making it challenging for casual motion designers to keep up with learning and using multiple (often feature-rich) software tools |
| 7 - | Motion graphic design is a collaborative activity that requires back-and-forth between designers and stakeholders using different tools. Motion designers desire support for in-context communication and collaboration within the tools that they use for design. |

Table 2: Key takeaways about motion designers' current processes and perceptions

A primary workaround for participants was visual content, such as low-fidelity storyboards, sample images from the Internet, or online videos that help them convey their intentions easier. As found in other works [50,56], visual communication is essential as not all stakeholders (e.g., clients) have the imagination to foresee the future of video from text-based descriptions:

*We have half an hour conversation, two days later I send them some images, and then if they say, "I love this one, but I hate this one." But can't explain why they hate the one or love another one, and the images are kind of similar. (P9)*

### 4.5 Key Takeaways

The key takeaways from this study (summarized in table 2) illustrate that all motion designers agreed that the language and terminology used in most tools were challenging to understand for users without a background in design or animation. The visual nature of motion graphics led motion designers to devise various visual communication strategies, some of which were not always successful. Furthermore, for many motion designers who worked remotely, online communities were essential in creating and learning by example. However, navigating the way around these communities and locating relevant content was another challenge for motion designers.

### 5 DISCUSSION

Our results indicate that despite the promise of motion graphics for informational domains [8,23,57], casual motion designers, who were domain experts in fields such as science or education, faced an uphill battle trying to create a motion graphics video. Although some domain experts had solicited professional motion graphics services, this is not a viable option in the long run because of high costs. Furthermore, the subject matter's nuances were often not accurately captured by professional designers, forcing domain experts to tinker with complex software on their own.

Our study has illustrated critical barriers for entering the motion graphics field (summarized in Table 2) shaped by the need for interdisciplinary skills, availability and a working knowledge of feature-rich expensive software applications, and continued access to a strong community of professionals. For casual motion designers, learning the role of different steps was understandably challenging as none of the current software tools integrate all stages of the process (pre-production, production, and post-production) in

one package. Furthermore, choosing a video style and interacting with feature-rich software applications were among other significant challenges. Although even professional motion designers were not immune to difficulties, such as labor-intensive animations, some had developed successful workarounds over time. However, even this group struggled to share their ideas, drafts, and creations with stakeholders.

While there is an opportunity for the HCI community to better support all motion designers, the need to understand casual motion designers who could benefit from these videos for their domain-specific needs is more urgent. Future initiatives can democratize creating motion graphics to empower more users to convey their ideas and complex concepts using this rich information-sharing medium. Similar challenges have been tackled in HCI for many years to close the gap between difficult technical tasks and content creation by casual users (e.g., complex domains and UX [14], supporting software programming participation [14], visualizations for everyone [43], or website creation [35].) We take inspiration from these efforts and offer future directions specific to lowering the barriers-to-entry for new motion designers.

### 5.1 Quick and Dirty Content Creation

In our study, we found out that one of the challenges of beginning with motion graphics for novices is the issue of a "blank canvas." Having no content to begin with deters novices' confidence. Future works can explore options for creating content "quick and dirty" to address that. Using libraries or premade content, icons, pictograms, automated text to image synthesis [56], or dynamic illustration [39], such tools can enable designers to turn their ideas into visual artifacts immediately. Furthermore, content-aware tools can synthesize users' ideas and offer relevant imagery from previously created content to inspire new videos. Such automation techniques can simplify the content creation process and increase beginners' confidence in the process. Doing so will enable those who are currently using low-level tools with limited motion graphics capabilities (e.g., slideshow tools) to leverage the benefits of using motion graphics while at the same time providing interfaces that simplify the content creation for more advanced users.

### 5.2 Contextualizing the Design Process

One of the main findings of this study was that new and casual motion designers faced difficulty in understanding the role of each of the pre-production and production steps in the broader context of their projects. For instance, they did not appreciate the value of

creating a shot-by-shot breakdown, storyboard, or designing a style frame before directly tinkering with a design or animation software. Furthermore, the need for multiple software applications to complete a project made the process rather complex and dispersed for casual motion designers. This area introduces an opportunity for HCI and visualization research to draw upon the successes of approaches used to enhance the design of interactive user interfaces [47,51]. Future work can explore techniques to support presenting each step of the process within a broader context of the entire project to help casual motion designers understand how different stages relate to one another. Such tools can allow users to navigate the distinct steps, relate the outcomes of the various steps in one software, and subsequently, better contextualize the process. The key goal here would be for beginner motion designers to not only better understand but also actively embrace the crucial pre-production steps of motion design.

### 5.3 Facilitating Content SaaS-ification

We learned that most casual motion designers were concerned with producing their videos as opposed to earning mastery in the trade, which was not surprising given that they already had day jobs in other domains. There is an opportunity for designing tools that could better support the "Software as a Service" approach. Such democratization and disintermediation have already been used in domains such as web development and 3D printing. In doing so, technical tasks are delegated to experts, and casual users focus on creating content. Some recent efforts in online motion graphics tools [3] have also begun working towards this strategy. Utilizing such a philosophy in motion graphics can enable casual motion designers to benefit from shared artwork, scripts, visual references, and pre-animated artworks to simplify the creation process. Pre-made content in such an environment can be adjusted to other users' specific demands on a shared platform. Doing so will eliminate the need for the tedious, time-consuming, or repetitive tasks for casual motion designers (e.g., animating). In such settings, the techniques of a community of experts can be harnessed to enable motion designers to focus on their messages.

### 5.4 Enhancing Learnability of Motion Graphics Tools

The sheer number of tools, menus, and user interfaces' complexity was a common challenge noted by motion designers. HCI has long focused on improving user interaction with feature-rich applications through appropriate user interface design or investigating approaches for improving software learnability [64,65]. Building on this work, there is an opportunity for future research to address the usability and learnability difficulties of current motion graphics tools. For example, one approach could be embedding interface divisions in which aspects of the software related to creating motion graphics are separated from other techniques such as video editing or compositing. Allowing further customization of the software tools using strategies such as user-controlled adaptable menus [48], drag and drop node-based systems, and encouraging an environment of sharing customized user interfaces can support and casual motion designers.

In our view, any or a combination of these approaches will enable individuals across many domains, such as education or science, to benefit from sharing their knowledge using motion graphics videos. Involving more individuals will also result in more creative ways to be developed and the tools to be utilized in unforeseeable ways and further support information sharing democratization.

### 6 LIMITATIONS

Although our study included perspectives from a broad range of motion designers, our focus was mainly on the processes and challenges of getting started in the motion graphics field. Future

work can include studies of motion design professionals and the challenges they face when working on more advanced projects. Moreover, our results showed that Adobe After Effects has a significant prominence in the industry, and our participants suggested that they do not have much to compare with it. Future work can consider other research methods, such as case studies, to contrast multiple software applications within a smaller organizational or team setting and even consider controlled studies for comparing different tool interventions and approaches.

### 7 CONCLUSIONS

Motion graphics were traditionally created by professional artists and mainly used in filmmaking or advertising productions. However, they are being used in various informational contexts, and there is an increased interest from domain experts to use motion graphics to explain complex concepts, such as in education. The issues of high costs encourage many individuals to become casual motion designers and take it upon themselves to create their own motion graphics videos. Our study is among the first to contribute empirical insights into motion designers' current practices and workflows. Motion designers face several challenges: from the initial training and onboarding to completing their first motion graphics video to using feature-rich software and communicating and collaborating with stakeholders. We have identified several opportunities for future work in HCI to build on this knowledge and innovate on tools and processes that can lower the barriers-to-entry to the field of motion graphics. By doing so, we can help casual designers participate in creating motion graphics and communicate complex informational concepts using this rich medium. We imagine a world where a domain expert (such as an educator) can leverage motion graphics videos as part of their daily routine with the same ease with which they can integrate slideshows.

### 8 ACKNOWLEDGMENTS

We thank the Natural Sciences and Engineering Research Council of Canada (NSERC) for funding this work.

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
