# OpenReview forum: "Challenges in Getting Started in Motion Graphic Design: Perspectives from Casual and Professional Motion Designers "
_graphicsinterface.org/Graphics_Interface/2021/Conference — GI 2021_

### Official Review · AnonReviewer3 · 2021-01-11
**Well-written paper exploring an interesting question of how to support casual motion designers. This reviewer has concerns about generalizability of findings, novelty of design recommendations, and some minor points on clarity of writing.**

**Rating:** 5
**Confidence:** 4

**Review:**

This paper presents the results of a qualitative interview study of the challenges and barriers casual and professional motion graphic designers experience when creating motion graphics videos. Based on the findings, design opportunities for HCI user-centered tool support are proposed.

The paper is generally well-written. The use of interviews is well-justified and the methodology section is explained with appropriate detail. Overall, the goal of supporting domain experts who are “casual motion designers” is an interesting direction for exploration. I do have some concerns regarding 1) the generalizability of findings, 2) novelty of design opportunities and its relation to previous research and 3) clarity of writing. I believe addressing these will strengthen the manuscript and increase the significance of its impact.

GENERALIZABILITY OF FINDINGS AND DESIGN IMPLICATIONS.
In the “Limitations” section, it is stated that “Adobe After Effects” has a significant prominence in the industry”, and participants “did not have much to compare it with”. Can the authors clarify what software systems the 19 participants used when creating motion design videos? At present, it is unclear whether these are general problems experienced causal motion designers, or whether the interview findings largely reflect the challenges of using Adobe After Effects? If it is the latter, the generalizability of findings and design implications may be problematic.

DESIGN OPPORTUNITIES.
Sections 5.1, 5.2, 5.3, 5.4 seem a bit high-level and vague.

The design opportunities are mostly framed with regards to what is lacking in commercial software. Can the authors also ground their design recommendations in existing academic research? There seems to be a lot of work in this area, but at present, the authors only address or mention this superficially. Could the authors contextualize their design recommendations in relation to current literature? (e.g. with regards to novelty / significance?)

I agree that there are challenges using feature-rich software tools but also wondered about the abundance of existing online tutorials and how or whether participants were able to utilize those. Is the barrier feature-rich software applications or is the barrier perhaps helping causal designers find appropriate, personalized resources to help them achieve their task while using feature-rich software?

Suggestions -
a)  I was surprised that none of the design opportunities discussed technological support for the creative storytelling challenges of creating motion graphics videos. The interview findings suggested that the creativity aspect may be a possible barrier, e.g. “blank canvas”, “the confidence to”, “getting inspired”, “creative freedom”.  This may be an interesting direction to explore in this manuscript or future work.
b) I was also surprised, given the interview findings, that technology support for building online communities, learning or a shared knowledge base was not proposed in the design opportunities section. Can the authors clarify why this was the case?

CLARITY OF WRITING.
From the introduction and related work section, I was not sure exactly who the study is about (casual designers, professional designers, or both). It wasn’t till Section 5 that clarified the authors’ focus on domain experts who undertake the task of creating motion graphics videos and who need to convey nuanced subject matter. I suggest the authors make the focus on domain experts clearer in the framing and introduction of the work. The authors may also consider introducing the paper with a scenario of a domain expert and their motivation/challenges/workflow/tasks, etc to paint a more vivid picture of who the user is and the barriers they experience.

MINOR.
- Figure 3 is confusing. It introduces new information and terms (e.g. “briefs) but these terms aren’t described in the text. Also, the caption talks about how casual designers skip “these steps” but the words “pre-production”, “production” and “post-production” are not present in the Figure, nor explained in the text.
- Missing period at end of first paragraph Section 3.1
- Missing colon page 5, end of paragraph “.... to create different content types”
- Section 4.5 should refer to Table 2.
- Section 4.4 seems more relevant to professionals, rather than casual designers.

---

### Official Review · AnonReviewer2 · 2021-01-13
**Insight into designers' workflows and challenges for making motion graphics videos**

**Rating:** 6
**Confidence:** 3

**Review:**

This paper presents the results from a series of interviews with 19 professional and hobbyist designers of motion graphics videos, presenting the results in a way that aims to highlight HCI opportunities for improving the accessibility of tools for creating such videos.

Overall, I find the quote-heavy insight into motion video designer's processes and challenges to be quite interesting and potentially useful to others in the area. This is the biggest strength of the paper.

The biggest weakness of the paper is the lack of a clear research thread that connects all the pieces throughout. The paper aims to improve accessibility to entry to creating these videos, and yet the up-front argument for the need isn't convincing. To be clear, I agree with the authors, but the argument as presented needs to be better grounded, e.g., in research telling us that people struggle with this. Points such as cost do not help, e.g., professional websites easily cost more than the figure listed, and yet simple websites are now very easy to create. Consider backing this better.

 - the RW seems to focus more on defending the importance of motion graphics and not situating the research goal of accessibility to new users. The end has a little on feature-rich software, but I feel this should be the focus.

- The selection of experts / experienced people to learn about how to support accessibility (instead of studying beginners, for example) is suspect and not well defended.

- The research questions leading into the semi-structured interview design and analysis strategy were not clear, and the ones hinted at (understand their struggles) are not defended from the perspective of supporting accessibility.

- as a result of this, the insights / outcomes for HCI at the end read quite generic and not that helpful unfortunately.

In my personal opinion, this paper would be a lot stronger if the authors worked on streamlining the argument, making it more concise (it's currently verbose), and focused more on the value of understanding experts' challenges. It would be a lot shorter and have more potential for impact.

(finally, small point: I did not find figure 2 helpful.)

In the end, for a GI paper I feel that this could be accepted based on the qualitative data as presented, although I worry about the potential for impact given the framing issues: readers may just look over it or ignore it based on the flaws listed.

---

### Official Review · AnonReviewer1 · 2021-01-13
**Challenges in Getting Started in Motion Graphic Design: Perspectives from Casual and Professional Motion Designers**

**Rating:** 6
**Confidence:** 3

**Review:**

The paper presents challenges faced by motion graphic designers based on an interview of 19 casual and professional motion designers working on a range of motion graphics projects. Their results reveal the difficulties of getting started in the field and the workarounds proposed by motion designers of varying expertise. Based on these findings, the authors identify opportunities for HCI
to lower entry barriers by designing user-centered.

The paper is relevant to the HCI community and fairly well-written. I am particularly impressed by the diversity of the interview participants with respect to the countries represented. However, the participants are majorly males (75%) and it seems like most of them are University students. That brings to question, the generalizability of the findings across gender at least.

Although only 5 females were involved in the interview compared to 14 males, it may be interesting to explore possible differences and similarities in barriers to entry, challenges, and proposed solutions across males and females. I would also like to see some reflection on the possible relationship between identified challenges and participants' training background. Perhaps, people trained in more technical backgrounds such as design and other IT related background may experience lower or different challenges compared to those trained in some social science backgrounds such as education and communication.

---

### Meta-Review · Area_Chair1 · 2021-01-14

**Recommendation:** Accept
**Confidence:** 3

**Metareview:**

The reviewers are all in agreement that this paper has some valuable contributions and strength: the work is relevant to the community, has pretty solid methodology, and has potential to be useful. However, it also some major flaws. The biggest seem to be the limited grounding in relevant academic literature (see the reviews), while one reviewer highlights the lack of analysis on the demographics (e.g., imbalanced gender, background).

In all, I am inclined to recommend acceptance for GI based on the insight and usefulness of the interview data presented, as I think this can be useful to others working in the area. I recommend the authors to carefully consider the quality reviewer feedback in updating their manuscript.

---

### Decision · Program_Chairs · 2021-01-16

Accept